# Exploring Trauma- and Violence-Informed Pregnancy Care for Karen Women of Refugee Background: A Community-Based Participatory Study

**DOI:** 10.3390/ijerph21030254

**Published:** 2024-02-22

**Authors:** Shadow Toke, Ignacio Correa-Velez, Elisha Riggs

**Affiliations:** 1Intergenerational Health, Murdoch Children’s Research Institute, Melbourne, VIC 3052, Australia; elisha.riggs@mcri.edu.au; 2Department of General Practice and Primary Care, The University of Melbourne, Melbourne, VIC 3010, Australia; 3School of Public Health and Social Work, Queensland University of Technology, Kelvin Grove, QLD 4059, Australia; ignacio.correavelez@qut.edu.au

**Keywords:** refugee health, trauma-informed care, trauma- and violence-informed care, pregnancy care, Karen women, trauma-informed interpreting

## Abstract

Women of refugee background experience poorer perinatal outcomes when compared to their local-born counterparts. Women of refugee background have often experienced trauma and are likely to encounter barriers to accessing health services in host countries which can exacerbate their recovery from trauma and contribute to poor health outcomes. Trauma- and violence-informed approaches to care offer opportunities to address barriers to pregnancy care which may, in turn, improve these poor outcomes. Trauma- and violence-informed care is a framework that acknowledges a person’s experiences of trauma, recognises its impact and symptoms, and works toward resisting re-traumatisation by integrating knowledge into practice. Despite this, trauma- and violence-informed care in maternity care settings has rarely been explored from the perspectives of women of refugee background. This study aimed to explore trauma- and violence-informed pregnancy care from the perspectives of Karen women of refugee background using Community-Based Participatory Research methods. The lead researcher is a Karen–Australian woman with lived refugee experience. A Community Advisory Group was formed to support the study. Semi-structured interviews were conducted with seven Karen women of refugee background who had recently had a baby in Western metropolitan Melbourne, Australia. The data were analysed using Reflexive Thematic Analysis. Karen women shared what they considered to be important elements of trauma- and violence-informed pregnancy care. Three major elements were identified: (1) care design and accessibility; (2) promoting choice and control; and (3) trauma-informed interpreting. The critical importance of the interpreter-mediated setting was highlighted as women reported that they may not experience trauma- and violence-informed maternity care if they cannot access an interpreter or their relationship with the interpreter is unsafe. This study offers critical insights regarding the elements of trauma- and violence-informed pregnancy care that are important to Karen women of refugee background.

## 1. Introduction

### 1.1. Background

Women of refugee background living in high-income countries are more likely to experience depression [1,2,3,4] during and after pregnancy, and have a higher risk of giving birth to a low birthweight infant, preterm birth, infant mortality, and congenital anomalies when compared to women who are born in those countries [5,6]. The Australian Pregnancy Care Guidelines recognise that social factors impacting refugee populations may later become barriers for women of a refugee background when attempting to access pregnancy care, and these factors may contribute to poor pregnancy outcomes [7]. Examples include migration factors such as a lack of knowledge of the host country’s health system; cultural factors including perceptions of care; socioeconomic status including lack of access to childcare and financial difficulties; a lack of social networks; transport and mobility problems; a lack of suitable resources including interpreters and translated information; health professionals’ lack of knowledge of cultural practices; poor communication and negative experiences with healthcare; and adverse childhood experiences [7,8]. It is important that these potential barriers to care are understood and overcome to improve the experiences of pregnancy care and subsequent outcomes for women of refugee background and their children.

#### 1.1.1. Trauma-Informed Framework

To improve women’s experiences of and access to antenatal care, trauma-informed care (TIC) offers a framework to overcome some of the factors impeding high-quality and culturally safe healthcare. TIC is a framework that acknowledges experiences of trauma of those involved in care, including health professionals, staff, individuals and families [9]. TIC recognises the impact of trauma and symptoms and aims to avoid re-traumatisation by integrating knowledge into practice [10]. A critique of TIC is its tendency to focus on past traumas, potentially overlooking the ongoing nature of violence [11]. Trauma- and violence-informed care (TVIC) builds upon the TIC framework, recognising the range of trauma and violence experiences individuals seeking services may have encountered. It also acknowledges potential biases and stereotypes held by service providers or organisations towards specific communities or groups [12]. TVIC argues that, given the prevalence of various forms of violence and the dynamics of stigma and discrimination, a non-judgmental, trauma- and violence-informed approach should be employed in all care interactions [11]. TVIC’s four key principles (see Figure 1) are as follows [12]:Understand structural and interpersonal experiences of trauma and violence and their impacts on peoples’ lives and behaviours;Create emotionally, culturally, and physically safe spaces for service users and providers;Foster opportunities for choice, collaboration, and connection;Provide strengths-based and capacity-building ways to support service users.

Although TVIC is noted as an emerging area of importance in developing a culturally responsive workforce, the concept or perceptions of TVIC have not been well explored among maternal populations of refugee background.

#### 1.1.2. Karen People of Refugee Background

By the end of 2022, there were 108.4 million people forcibly displaced worldwide due to persecution, conflict, violence, and human rights violations [14]. In recent years, Australia has resettled thousands of refugees as part of its Humanitarian Program. For example, in 2021–2022, Australia granted 13,307 resettlement visas as part of their offshore Humanitarian Program, of which, 14.2% were from Myanmar, and 7.5% were vulnerable women and children [15]. According to Australia’s most recent census reporting (2021), there are 13,602 Karen people of refugee background currently residing in Australia, of which, 7165 are in Victoria (Victoria is a state located in southeastern Australia) [16]. Karen (emphasis on the second syllable—“Ka-REN”) people are one of the largest cultural groups from Myanmar (Burma)—there’s approximately 7 million Karen people living in Burma [17]. Approximately half a million Thai–Karen people trace their ancestral roots to Thailand, with smaller Karen communities residing in India and various Southeast Asian countries [17]. Thailand hosts around 140,000 Karen refugees in camps, while approximately 50,000 Karen refugees have found resettlement in the United States, Canada, Australia, and select European nations [17]. Karen people have their own language (S’gaw Karen being one of the major dialects) and distinct culture and have been engaged in a long-standing struggle for autonomy and independence within Burma. Karen people have faced human rights abuses, including forced labour, genocide, persecution for their religious and political views, forced relocation, and military attacks. These abuses have caused thousands of people, including women and young children, to flee to neighbouring countries and seek refugee status.

The 1951 Convention defines a refugee as “an individual who has been forced to leave their country due to fear of being targeted based on their race, religion, nationality, membership in a specific social group, or political beliefs, and is either unable or unwilling, because of this fear, to return to their country” [18]. In this study, we use the term “refugee background” to encompass a broader spectrum, acknowledging that someone may have a refugee or refugee-like background without meeting the legal obligations of the term in the Refugee Convention in a signatory country [19]. It is important to recognise that not all those who arrive in Australia possess officially recognised refugee status, and acknowledging these diverse pathways is essential to understanding the multifaceted journeys individuals of refugee background take to reach Australia. Furthermore, intergenerational trauma can impact not only the first generation of refugees but also the second generation, with consequences that can transcend generations, influencing the well-being and life experiences of individuals from refugee backgrounds. Incorporating this intergenerational dimension enables a more comprehensive understanding of the intricacies and distinct challenges faced by these communities, particularly in the context of women with refugee experiences or immediate family members with such experiences [19].

Karen women have reported experiencing various types of trauma, including primary and secondary torture, and war trauma [20]. Primary torture involves causing severe physical or mental pain for purposes such as obtaining information or punishment [21]. Secondary torture is inflicted on the close contacts or family members of an individual [22]. War trauma encompasses psychological distress from exposure to danger, witnessing harm, or learning about extreme events in war [23]. These traumatic experiences can become deeply embedded within individuals and may persist throughout their lifetime, extending even through pregnancy [8]. Women of refugee background who have resettled in a host country may continue to experience trauma, discrimination, and marginalisation, including cultural and language barriers, and stigma about their religion, culture, or refugee status which further hinders access to health and social services [24,25]. Experiences of trauma affect the physical and mental wellbeing of women of refugee background, as well as their children. Evidence shows that adverse childhood experiences, such as war trauma, heightens the risk of future pregnancy complications and adverse outcomes, including gestational diabetes mellitus, antenatal depression, low birthweight in offspring, and preterm delivery [8]. As a result, they often experience inequalities in accessing health services in host countries which further exacerbates their recovery from trauma and can lead to poor health outcomes [26].

This study aimed to explore Karen women’s perspectives of trauma- and violence-informed pregnancy care in Melbourne, Australia.

### 1.2. Positionality

The lead researcher (ST) identifies as Karen and of refugee background with experiences of living and growing up in both Australia and a refugee camp in Mae La, Thailand, as well as being a professional interpreter and having several years of experience as a Research Assistant working in the area of refugee maternal and child health. ST assumed the roles as a student researcher for this Honours student project—drawing on her lived experiences with both cultures. By leveraging her experiences as a bicultural Karen–Australian woman when interacting with the Karen community, with which she shared a lived experience, she was able to apply cultural knowledge and understandings, language skills, and social networks within the Karen community to support engagement [27]. Furthermore, the lead researcher’s deep-rooted passion for collaborating with her community and her desire to serve as an advocate inspired her to initiate a study within her own community using the Community-Based Participatory Research methodology.

## 2. Methods

### 2.1. Methodology

Community-Based Participatory Research (CBPR) is an approach that aims to support efforts to improve health equity by emphasising the importance of involving community members at each stage of the research process including the design, implementation, and evaluation [19,28,29]. CBPR values the input of community researchers and community members to address social and health inequities that people of refugee background often experience [30]. It is recognised that the representation of ethnic minorities and bicultural academics in the research team is extremely important for cultural understanding [31]. The principle behind doing research with community as an ‘insider’ researcher who shares lived experiences with the participants in this study is that it may help to mitigate cultural and linguistic differences that are present between the research team (of non-Karen background) and participants from the Karen community [32].

### 2.2. Community Advisory Group

Establishing a Community Advisory Group (CAG) is a means of engaging the community in a culturally safe manner [33]. The use of advisory groups promotes equity within the community, for example, it promotes meaningful leadership in the community; builds community and individual capacity; and promotes social networks and partnerships with services [19]. A CAG was formed from the outset of this study, and the community advisors were consulted throughout the design and conduct of the study, and contributed to framing the research findings in a culturally safe and trauma-informed way. Figure 2 demonstrates the structure of the research team, where the CAG was situated in the study, and how they engaged with the research team.

In July 2021, the lead researcher [ST] leveraged her Karen community networks to identify suitable advisors among elders and recent mothers living in Melton in the outer Western metropolitan Melbourne, situated 44 km from Melbourne city, and Wyndham, situated to the southwest of Melbourne city, approximately 30–40 km from Melbourne city. The Melton and Wyndham regions exhibit a low socioeconomic status, significant diversity, and are key areas of recent refugee resettlement. The resulting CAG included six members initially, but three withdrew due to COVID-19 challenges (e.g., pressures from home schooling their children). The three advisors that continued to support the project were all well connected with their communities; one advisor was a community elder and leader from Wyndham, and the other two advisors were mothers with experiences of maternity care in Australia and both were living in the Melton region. The advisors together with ST, co-produced an agreement letter outlining the research project’s background, objectives, what a community advisory group is, and the processes of working together, including expectations of the community advisors, the researchers involved, and the project stages. Examples of agreements included creating a safe and confidential space for everyone to share their thoughts; communicating concerns and availabilities, objectives of meetings, and project timelines; and being transparent about decision-making processes.

In collaboration with the community advisors, we identified culturally appropriate methods for involving Karen women. For example, one-on-one interviews conducted by ST were deemed to be acceptable and the preferred method because women would feel they could confidently and comfortably share their experiences in their preferred language. The advisors recommended that the project use clear, simple language, avoiding jargon to address cultural and language barriers. Any decisions made by the research team were based on the community advisors’ advice and feedback. These decision-making processes were recorded by ST.

All meetings were held over Zoom due to COVID-19 restrictions to ‘work from home’. ST and the advisors met twice for two hours as a group and the advisors were reimbursed with an AUD 50 supermarket gift voucher for each meeting. ST contacted the advisors in between the meetings by phone when their input was required.

### 2.3. Participants

#### 2.3.1. Inclusion Criteria

S’gaw Karen-speaking women over the age of 18 at the time of recruitment who had received perinatal care in Melbourne, Australia, in the 24 months prior to the interview were eligible to participate.

#### 2.3.2. Sample

As is typical in exploratory qualitative research, we aimed to recruit a small sample of 6–8 women [34]. We arrived at the decision that utilising a modest sample size was not only feasible but also necessary, primarily due to the limitations posed by various factors such as the limited timeframe of the Honours student’s timelines, financial considerations, and the prevailing circumstances in the community, including the ongoing implications of the COVID-19 pandemic. This approach ensured the feasibility and integrity of the research while optimising the available resources.

#### 2.3.3. Recruitment

Recruitment occurred between late August to mid-September 2021 and resumed in June 2022. ST recruited the participants through convenience and snowball sampling via the community networks of the community advisors and promotion on social media (Facebook, Messenger) [34,35,36]. As the Karen community is small, participants were selected based on who was available at the time of recruitment (convenience sampling) and participants were asked if they knew anyone else who might want to participate (snowball sampling) [34].

#### 2.3.4. Consent

Verbal consent was used, a process that is well established by the Refugee and Migrant Health Research Program, Intergenerational Health Murdoch Children’s Research Institute (MCRI), for working with refugee background communities. ST read the Participant Information Sheet aloud in Karen, and asked a series of questions to confirm the woman’s understanding of the project and willingness to participate. ST documented the participants’ verbal consent, and signed the electronic REDCap consent form on their behalf. The Royal Children’s Hospital HREC approved this process of obtaining verbal consent, allowing ST to sign the consent form, based on her professional interpreter qualifications and language proficiency.

Participants were provided with an AUD 50 supermarket gift voucher at the end of the interview in recognition of their time.

### 2.4. Data Collection

Semi-structured individual interviews were used to collect data, as this method is the most ideal for research questions exploring understandings, experiences, and perceptions of participants in a confidential manner [36]. This approach was supported by the community advisors. The advisors were asked: What is the main concern for Karen women and their families when attending pregnancy appointments? Based on their answers, an interview guide was developed and piloted with two Karen women to assess cultural safety and comprehension [36].

The interview began with an open-ended question prompting women to share their pregnancy experience in its entirety, asking, “Tell us about your pregnancy experience, from beginning to end, what was it like?” In instances where women found it challenging to articulate their experiences, prompt questions were employed to assist them in narrating their story. Examples of the prompt questions asked were as follows:What was your pregnancy care like?What did/didn’t you like about it? Why/not?Is there something that you wished had been part of your pregnancy care?What would have been like if that had been part of your care?When you were pregnant, what did you feel that you needed?What was it like using interpreter services?What do you think every Karen family needs in pregnancy care?Was there anything you found difficult when attending your appointments?

The concept of trauma and its effects on women’s maternity experiences and outcomes is not easily translatable in S’gaw Karen, and so gauging women’s understandings and reflection of their maternity experiences through the lens of trauma was not straightforward. ST had consulted with the community advisors on whether it would be appropriate to discuss trauma with women. Their advice was (1) keep the interview simple; (2) clarify why the interview questions were being asked; and (3) make the questions straightforward and easy to understand.

After the initial interviews were conducted, the Community Advisory Group advised ST to introduce the concept of trauma- and violence-informed care prior to asking the women questions about their experiences. ST shared how people’s refugee experiences and prior exposure to trauma could influence their current maternity care. Subsequent interviews were approached in this way to hone in on women’s experiences in light of trauma-informed approaches.

Due to COVID-19 restrictions, the interviews were conducted via phone or Zoom depending on the woman’s preference. The interviews were audio recorded using an external audio device with the women’s permission.

### 2.5. Data Analysis

The recorded audio of the interviews were all in S’gaw Karen, which ST transcribed and translated at the same time into English. When ST translated into English, ST included different English variations of the meaning or concepts and corresponded it to the original word in Karen. This was important to do as some words or concepts cannot be directly translated. Keeping the corresponding word in its original spoken language was important context for ST to reflect on when performing the data analysis [36]. The data analysis started after the first interview and continued during the data collection.

ST conducted a reflexive thematic analysis (TA) using interview transcripts, fieldnotes, and audio recordings [37]. In the fieldnotes, ST documented her reflections on the interviews, any impressions she had during the interviews, and any relevant cultural context as to why participants may be responding the way they were. The prompt questions were not employed for organising the data. Instead, the data analysis focused on the most prevalent experiences women shared, with themes named to accurately encapsulate these experiences. Reflexive Thematic Analysis is an approach to data analysis that centralises the subjectivity and reflexivity of the researcher to TA [38]. Braun and Clark describe themes as analytic work that has been created by the researcher through an active process of work and reflection on the intersection of data, analytic process, and subjectivity [38]. Throughout the data analysis, ST reflected on her position as a Karen–Australian student researcher, and as a Karen woman of refugee background with lived experiences as a refugee. The lead researcher, ST, was careful in interpreting the meaning of women’s responses—ST continuously reflected upon the cultures and experiences of Karen people in an attempt to gather meaning from the women’s responses. The lead researcher conducted the data analysis, completing the data coding and categorisation manually. ST did not utilise any data analysis software.

Elements of trauma- and violence-informed care were identified from the women’s experiences by recognising that what women liked or what made them feel happy/safe/good/comfortable is what they would consider to be trauma- and violence-informed care that is safe and avoids re-traumatisation. Women were frequently using the word “မုာ်” (mu-) or “တမုာ်ဘၣ်” (dt-mu-bah) to describe their experiences of care. “မုာ်” (mu-) can be translated interchangeably as happy, safe, good, and/or comfortable. “တမုာ်ဘၣ်” (dt-mu-bah) means “not happy, safe, good, and/or comfortable. Women also sometimes described the feelings they experience during pregnancy care as feeling “သဆံး”, which directly translates to “small hearted/spirit”. This word is often used in Karen to describe a negative feeling that can interchangeably translate into English as feeling unsafe, small, insignificant, inferior, and/or not confident. During the data analysis, ST carefully reflected on the emotions the women were trying to convey and chose to use some English words to convey the meaning. But, it is important to note that because language is intricate, sometimes it was necessary to use all the interchangeable words in English to fully grasp the experiences of the women.

During the data analysis, ST consulted the community advisors to ensure the initial findings were aligned with what they expected to hear from the community; in this way, the advisors validated the preliminary findings. ST invited the CAG to assist with making sense of the results from the draft themes [35]. Seeking to validate and refine this understanding, ST shared with them her exposition of the women’s stories and narratives. ST asked if what she had concluded so far was an accurate reflection of the experiences of women and their families within the wider community. ST recognised the depth of insight that the CAG brought to the study and entrusted them with the task of critiquing and enriching the analysis. By involving people who have a close connection to the community, a space was created where making sense of the data was not confined to the researchers alone but also included people who have a deep understanding of the lived reality being explored. From this meeting, it was determined that to consider what trauma- and violence-informed care is for Karen women, the language, culture, and lived experiences of the people should be considered.

### 2.6. Ethics

Ethics approval was obtained from the Royal Children’s Hospital Human Research Ethics Committee (HREC74463).

## 3. Results

In total, ST interviewed seven Karen women of refugee background residing in outer Western and Southwest Metropolitan Melbourne, Victoria, Australia. Five women were born in Burma and two were born in Thailand, and all self-identified as belonging to the Karen ethnicity and received perinatal care in outer Western and Southwest Metropolitan Melbourne in the 24 months prior to their interview. This means that all the women had their baby during the COVID-19 pandemic. All participants required interpreting services during their pregnancy care and were cared for in a standard model of public maternity care—all visits were in public hospital with a midwife as per the hospital schedule for visits. The women were aged between 20 and 40 years old at the time of their interview and had an average of two children each, with at least one of these children born in Australia. The women’s year of arrival to Australia ranged from 2012 to 2018, and they had been in Australia for an average of 4.5 years at the time of the interview.

Three key elements of trauma- and violence-informed pregnancy care were identified—what makes women feel happy and safe during their experiences of pregnancy care: (1) care design and accessibility; (2) promoting choice and control; and (3) trauma-informed interpreting.

### 3.1. Care Design and Accessibility

#### 3.1.1. Accessing and Navigating the System

For the Karen women, trauma- and violence-informed care means having access to additional support to attend their appointments. The women shared that it was difficult for them to go to their appointments because they were unfamiliar with how to physically get there and could only travel by public transportation or by relying on other people to take them. Because of this challenge, they indicated that it was important for their healthcare to be easily accessible. Additionally, if problems getting there on the day were encountered, it was important for women to be able to easily reschedule or cancel their appointment, even if their English proficiency is limited.

“*[my husband] can speak a little bit here and there, so I asked him to let [the hospital] know we cannot make it to our appointment and that we were lost. He just spoke however he could, maybe they could understand him.*”
*—Participant 2*


#### 3.1.2. Support Persons

The women felt happy and safe when service providers and health professionals included their support people as part of their care. Where the women felt their care providers could not offer additional support in accessing services, they felt it was their responsibility to organise their friends and family members for transportation, navigating healthcare services, and language support. The women identified the importance of having a sense of community, having “good friends” and “good health professionals” to help and support them during pregnancy by helping them get to appointments.

The women’s experiences of maternity care were most affected by the COVID-19 pandemic as they were instructed to attend their appointments alone due to hospital restrictions. The women shared that their fear and anxiety increased when they were denied the ability to have a support person with them when attending pregnancy care.


*Because we feel alone. Everybody would leave you… You’re just there aimlessly, I feel like I end up with more mental illness. Because you’re alone.*

*—Participant 5*


The fear and anxiety that the women felt when attending their pregnancy care alone, especially at the first appointment, was exacerbated by the lack of language support and the hostile treatment received from those they had to interact with at the hospital.

“*Some of them looked unfriendly and of course we are scared of them. We already cannot speak their language, of course we feel inferior.*”
*—Participant 5*


The women felt confident and safe when attending care with a support person. The women’s support person helped them with getting to the appointment, navigating their way around the hospital, and sometimes providing language support and support in accessing information and optimal care.

“*So when I have my relatives in the appointment with me they can help me by asking questions and thinking of what to say or talk about. What we can’t think of, they can help think for us and ask for us.*”
*—Participant 7*


One woman, who began her pregnancy care just before the COVID-19 restrictions started, gave an example of how health professionals and services can enable trauma- and violence-informed care by welcoming women’s support person to their care.

“*… If they make an appointment with me, I will have to take a friend. And they will give me an appointment time that my friend is also available for. That’s how they do it for me.*”
*—Participant 3*


### 3.2. Promoting Choice and Control

#### 3.2.1. Roles and Responsibilities of the Health Professionals

The women felt they knew little about their health professionals’ roles and responsibilities in providing pregnancy care. Not knowing what to expect was a common experience that the participants shared; the women described that it made them feel unsafe and lost. In addition, the women did not feel safe/comfortable to question their health professionals’ advice and medical instructions. Most women felt that their health professionals were “*just doing their job*” and would often go along with whatever the health professional had organised because the health professionals were viewed as superior and in charge of their care.

“*I thought because they are care providers they will understand and know more than us.*”
*—Participant 6*


The women felt uncertain about what to expect from their appointments and whether there were any specific preparations they needed to make. The women valued their health professionals explicitly communicating to them that they had the freedom to ask for help or pose questions whenever needed. When the women did not understand their health professionals’ roles and responsibilities, they felt that it was their responsibility to find someone to support them with interpreting, and to seek out the support and information that they needed. When the women did not get the information or the support they needed, they felt that it was their fault for not asking or for not knowing how to access it.

“*It was my fault for not asking them.*”
*—Participant 2*


#### 3.2.2. Women Knowing Their Rights

In addition to understanding their health professionals’ role, the women identified the importance of being explicitly informed about their healthcare rights. Beyond comprehending the roles of their health professionals, the women recognised the significance of receiving clear and direct information regarding their rights related to healthcare. Often, the women felt they did not know who to ask if they had questions or know where to go to ask for help. The women wanted to know their rights and responsibilities regarding pregnancy care in order to avoid feeling small hearted/spirit, inferior or unsafe. A sense of predictability and what to expect from their pregnancy care supports women to have a sense of control.

“*I don’t know, I want to be able to speak a lot but I don’t know what to say about what we need. I don’t know. I guess we need their help and support…They told me everything I wanted to know. And the things that I didn’t, well I was new and I didn’t know how to ask questions so they told me things, asked me to do things. That’s it.*”
*—Participant 3*


#### 3.2.3. Comfort in Good Care

Good pregnancy care to the Karen women is when they felt comfortable with their care providers, one who they felt treated them with care and spoke to them nicely. The women felt a sense of security in care when they felt comfortable with their care provider and felt they could tell their health professionals about any concerns, worries, or wants. The women experienced a feeling of safety and comfort in their healthcare experiences when they were able to openly communicate with their health professionals about any issues, anxieties, or preferences they had.

“*If a [pregnant woman] feels like their care provider is good, it’s someone who speaks nicely. I feel like when you get a care provider like that, your pain and problems are already half gone.*”
*—Participant 4*


#### 3.2.4. When Women Felt Unsafe

Some of the women felt unsafe in their appointments, and in response, they purposely delayed access to care to avoid seeing certain health professionals.

“*The next time I got pregnant, I knew about it and purposely did not go for a check… It was like I was left [feeling] scared… When I had my daughter, I went for a check after 12 weeks (of pregnancy).*”
*—Participant 4*


In S’gaw Karen, the concept of “trauma” or “traumatised” does not have a direct translatable word; thus, when the participants mentioned they were “*left [feeling] scared*”, it can be understood as feeling traumatised.

### 3.3. Trauma-Informed Interpreting

#### 3.3.1. When There Is No Interpreter

The women described interpreters as both potential barriers or facilitators of trauma- and violence-informed pregnancy care. The participants shared that they did not always have an interpreter at their appointment despite needing one. When there is no interpreter, the interaction and communication between the woman and the health professional is severely compromised. The women felt frustrated because they wanted to express a concern to their health professional but there was no one that could interpret for them.

“*Something important, I wanted to tell my health professional something, and there’s no one to interpret for me and I couldn’t tell her. I did not feel good.*”
*—Participant 2*


The women did not always know why interpreters were not at their appointments. The women shared that sometimes despite making the difficult journey to their care, they’d have to go back another day because there was not an interpreter provided at their appointment.

“*Sometimes when the interpreters don’t come, we have to go back another day.*”
*—Participant 1*


#### 3.3.2. Even When There Is an Interpreter

Sometimes, despite having an interpreter, the women still experienced challenges in accessing quality care. Even when interpreters were present, the women did not always experience trauma- and violence-informed care. The women shared the importance of having competent interpreting in care and not just the mere presence of an interpreter. How the women felt about their care was influenced by the way interpreters interacted with them. Thus, it was important for the women to feel they had a good connection with their interpreter. When interpreters participated in creating a safe space, the women felt happy during their care.

“*If [the interpreter’s] tone is nice/friendly then it makes you feel good.*”
*—Participant 5*


The women spoke about instances in which they felt that the interpreter was growing impatient towards them, displaying frustration and speaking sharply when the women asked for repetitions or posed questions. The women shared that in such situations, they did not feel safe enough to seek clarifications or ask questions. Consequently, they faced difficulties in expressing their feelings comfortably, resulting in a sense of unhappiness and a lack of safety while receiving care.

“*If they [interpreter] are being like that then I don’t want to ask questions anymore.*”
*—Participant 7*


One woman shared that she felt concerned about the interpreter misinterpreting what she said because she felt they could not understand her because of the differences in how they speak Karen. And she felt she could not ask for information to be repeated or clarified because the interpreter would get angry with her.

“*The health professional asked me whether I understood the interpreter and I said no, not really, he didn’t accurately interpret what you’re saying and what I’m saying. She said to me: because you couldn’t understand him, I will not call him again. There won’t be any difficult words for you to understand. Your husband will be able to understand also… I felt good.*”
*—Participant 4*


This woman’s experience suggests that in cases where the interpreter is not practicing trauma-informed interpreting, health professionals can enhance women’s care experiences positively by assessing their satisfaction with the interpreter and responding appropriately to their feedback.

“*They got an interpreter for me, but the interpreter just talked nonsense, the providers that were looking after me they changed to a new interpreter for me.*”
*—Participant 6*


Some women, who spoke and understood a little English, shared that they would rather have their health professionals speak slowly and clearly to them in English than wait to find another interpreter, whom they worried may or may not be good. However, they would feel more confident with the right interpreter.

“*Well I told them, if you don’t have an interpreter for me, talk to me slowly and that’s what they did so that was fortunate. But I just felt more assured if I had an interpreter, it would be more accurate.*”
*—Participant 5*


One woman shared that at times she felt confused with an overload of information because the health professional and the interpreter did not work together well to take the time to make sure she understood what she was consenting to. She felt clueless and did not understand the information even with an interpreter.

“*And some providers, the interpreter haven’t finished telling us all the information but the providers would stop the interpreter and interrupt them and then they say something to us later and we don’t understand everything. I think it will be a bit harder for those who do not understand it at all.*”
*—Participant 6*


“*Yes, they interpreted… I still think about it… They asked a lot of questions and I had to sign a lot of things… I didn’t think anything of it at first… I thought it was important… they said I can sign if I wanted but I don’t have to… but I only arrived (in the country) recently so I didn’t understand…*”
*—Participant 1*


#### 3.3.3. The Right Match

The women felt confident that they were getting the care and information they needed when they perceived that the interpreter was explaining everything accurately. When asked “what is a good interpreter?”, one woman said

“*for example, a good one is… this [interpreter] spoke to me very well, he speak to me clearly, and he was not fast, he spoke clearly, if you don’t understand he will repeat it for you, and he will check with the health professional as well.*”
*—Participant 7*


Based on this woman’s definition of a ‘good interpreter’, providing trauma-informed interpreting in a maternity care setting remains feasible even with a male interpreter, as long as the interpreter conducts themselves with courtesy, professionalism, and ensures that the woman comprehends the information. Nevertheless, it is noteworthy that the women still conveyed a preference for female interpreters, enabling them to discuss women’s health more comfortably.

## 4. Discussion

This study explored with seven Karen women of refugee background living in Melton and Wyndham in Melbourne, Australia, what they considered to be trauma- and violence-informed pregnancy care. A Community-Based Participatory Research design was used to guide this study with a Community Advisory Group involved at all stages of the research process. In addition to strengthening the understanding of what it takes for women to feel happy and safe in care, this study identified unique opportunities to avoid re-traumatisation for Karen women when accessing pregnancy care.

### 4.1. Happy and Safe

This study found that women feel happy and safe in care when their care is accessible, they have access to trauma-informed interpreting, and their support persons are integrated into their care. The Karen community, like many refugees, often lives with constant fear and uncertainty about the future [39]. Sometimes, pregnant women from the Karen community were subjected to public torture by the military as a means to instil fear and shame [39]. This experience adds to the vulnerability of refugee women who often face violence during migration, leading to high rates of post-traumatic stress disorder [39]. The research indicates that post-migration challenges are the strongest predictor of traumatisation, PTSD, anxiety, and depression [40]. The women in this study also expressed anxiety and fear, especially when facing the unknowns of the Australian maternity care system. This fearful experience can be re-traumatising for them. To access and feel safe and happy in pregnancy care, Karen women require language and practical support. When services and health professionals incorporate women’s support people and facilitate access to trauma-informed interpreting, this supports predictability, consistency, and cultural safety in pregnancy care. This study extends past work such as that by Au and colleagues (2019), who performed a systematic review of 35 articles exploring refugee perceptions of the Australian healthcare system [26]. Au and colleagues (2019) identified the importance of choice and control in efforts to prevent re-traumatisation in care for people of refugee background [26], including those accessing maternity care as identified by Pangas and colleagues (2019) [41]. It is essential for health professionals working with people of refugee background to actively work towards preventing re-traumatisation [42], whether inflicted by a spouse or partner [43], or unintentionally by the provider through secondary victimisation. Secondary victimisation refers to actions and attitudes of service providers that may blame the victim and lack sensitivity [44]. Thus, trauma- and violence-informed pregnancy care necessitates a commitment from services and health professionals to promote access to appropriate interpreter services and integrate women’s chosen support people into their care.

### 4.2. Not Knowing, Power Imbalance, and Re-Traumatisation

This study demonstrated women’s experiences of power imbalance and re-traumatisation by not having a clear understanding of their health professionals’ roles or what pregnancy care entails. As a result of not knowing their health rights, women may blame themselves when they are unable to obtain the healthcare and information they require, believing that it is their responsibility to obtain it. Pangas and colleagues (2019) conducted a meta-ethnography of 25 papers that explored the experiences of refugee women as they navigate motherhood and access maternity care in high-income countries [41]. They reported that self-blame and self-reliance can be extended to women’s mental wellbeing, in that they feel it is their responsibility to “control [their] heart” [41]. Not knowing can create a sense of lack of control which has the potential to remind them of how they felt before resettlement. A lack of knowledge and familiarity with healthcare services in host countries can contribute to power imbalances between women and health professionals [41], which may exacerbate disempowerment, distress, and fear, with the potential to cause re-traumatisation [26]. These power imbalances may be reminiscent of past experiences. Karen women have been noted as being particularly at risk of gender-based violence, including violence perpetrated by those in authority, such as the Burmese and Thai militaries [45].

Increasing choice and control in the provision of health care can help to prevent re-traumatisation [9,41,46,47]. Not knowing can be addressed by explicitly explaining to women their healthcare rights and the role and responsibilities of health professionals [26]. Doing so shares the responsibility and power in pregnancy care between women and health professionals by clearly highlighting how health professionals can offer accessible and appropriate care to women during pregnancy. From this, provided that there are no other barriers, women may feel safe, happy, and empowered with a sense of control and choice over their pregnancy care. The women in this study provided examples of elements of trauma- and violence-informed care that support choice and control. This includes having knowledge of health professionals’ roles and responsibilities, understanding their healthcare rights, and being able to communicate with health professionals effectively and consistently via trauma-informed interpreting services. Wathen and Varcoe’s examples of structural violence [48] and the potential for re-traumatisation when interpreters are not available are evident in this study. Social arrangements rooted in racism or classism expose individuals and minority groups to “structural violence”, a repetitive and often normalised form of harm in society [49]. Refugee background women who have resettled in host countries experience persistent discrimination, marginalisation, cultural and linguistic obstacles, as well as stigma associated with their religion, culture, or refugee status. These challenges hinder their ability to access crucial health and social services [24,25].

This study has identified that even when a woman has a ‘good health professional’, is aware of her healthcare rights and the role of her health professionals, and efforts have been made to address power imbalances in care, she may not experience trauma- and violence-informed care if she cannot access an interpreter or her relationship with the interpreter is unsafe.

### 4.3. Trauma-Informed Interpreting

Trauma-informed interpreting has been noted as a “young yet vital specialisation” [50]. The intent of trauma-informed interpreting, as defined by Bancroft, aligns with the broader goals of TIC, which aims to integrate research on trauma into professional practice [50]. However, the concept of trauma-informed interpreting has not yet been applied to maternity care.

The Karen language is diverse even among its native speakers; accents, slang words, and ways of expression vary among different Karen villages. Despite language differences, the right interpreter, one who practices trauma-informed interpreting, is one that can relay messages in a clear and simple way so that women and health professionals can understand each other. Interpreters have an important role to play in providing trauma- and violence-informed care for women during their pregnancy by aiding smooth communication and establishing trust and a safe setting. For women, establishing trust is important to enable trauma-informed interpreting [50]. Continuity of trauma-informed interpreting may support non-verbal and verbal communication between women and their interpreters [50]. Thus, this study emphasises the importance of having a gentle, safe, and trauma- and violence-informed approach when interpreting [51].

Refugee background communities experience a number of barriers to interpreting services within the Australian healthcare system [26]. The meta-ethnography by Pangas and colleagues found that these barriers persist within maternity services; however, the discussion within this review focused on barriers to access to interpreters without acknowledgement of the importance of a positive relationship with the interpreter when present [41]. The relationship between the woman and interpreter is an overlooked element of trauma- and violence-informed pregnancy care. The Karen women in this study explained that when they could not access an interpreter or when their relationship with the interpreter was poor, they could not access the care they needed or share their concerns with their health professionals. This led the women to feel unhappy and unsafe in care. When interpreters are not involved in creating a safe trauma- and violence-informed space, the women stopped engaging in their care by not asking questions or clarification. When women stop engaging in their care, it means they are not receiving or comprehending important health information that they might need.

## 5. Strengths and Limitations

This study has identified elements integral to trauma- and violence-informed pregnancy care for Karen women. At the present time, there are no existing studies that focus on trauma- and violence-informed pregnancy care for Karen women of refugee background, so this study is among the first to fill this gap.

The research team acknowledges the limited number of advisors and participants engaged in this project. Given that this study was a small student project, it was beyond the scope to fully investigate how prior trauma can inform current experiences and how certain hardships may be compounded by poor pregnancy care.

Nevertheless, the team deemed it appropriate given the modest size of the involved community, thus rendering a large advisory group unnecessary. The chosen advisors possessed profound connections within their respective localities. Despite the apparent modesty in numbers, substantial efforts were devoted to fostering community engagement in a manner that ensured the safety and trustworthiness of the participants, researcher, and generation of data. The team placed paramount importance on ensuring safe community involvement rather than having a large sample of participants and advisors. Furthermore, after conducting seven interviews, it became evident that the data were reaching a saturation point, as the participants consistently shared similar experiences.

While the study has provided new insights, it cannot definitively establish trauma- and violence-informed pregnancy care for women of refugee backgrounds beyond the Karen community in the west of Melbourne. The study exclusively focused on Karen women, highlighting that experiences and preferences for pregnancy care can be significantly shaped by cultural and linguistic factors. Therefore, it is reasonable to expect that other cultural groups or distinct Karen ethnic groups may have slightly different experiences with Australian healthcare. This research is limited in that it solely examines the topic from one perspective. The authors suggest that future studies should involve additional stakeholders in maternity care, such as interpreters and maternity care providers.

The CBPR approach adopted in this study—led by a Karen–Australian woman—is a key strength of this study, in particular the incorporation of a Community Advisory Group from the outset. The CAG supported cultural safety and ensured that the study was responsive to community needs. The research team carefully thought through critical issues that may arise, and collaborated efforts to address language barriers and ensure that all participants were well informed about the project and the scope of their involvement. To protect their privacy and ensure cultural safety of the small number of participants, only limited demographic and personal information was collected. The student researcher, herself an Australian–Karen woman, was able to form and facilitate the CAG, conduct interviews in the women’s preferred language, and translate and transcribe the interviews into English. Cultural and language insights were integral to the data analysis process and have contributed to the quality and nuance of the findings presented in this paper, as seen in attempts to honour the ways in which women used Karen words to explain their experiences. The research findings will be communicated to the participants and the Karen community.

## 6. Conclusions

This study aimed to explore what trauma- and violence-informed pregnancy care is for Karen women of refugee background. Community-Based Participatory Research methods were adopted, including the use of a Community Advisory Group and the insights informed the student community researcher. With a commitment to culturally safe and community informed research practices, this study was able to identify that care design and accessibility, choice and control, and trauma-informed interpreting services are crucial elements of trauma- and violence-informed pregnancy care for Karen women. This knowledge presents important opportunities to address health inequities and prevent re-traumatisation for Karen women accessing pregnancy care.

## Figures and Tables

**Figure 1 ijerph-21-00254-f001:**
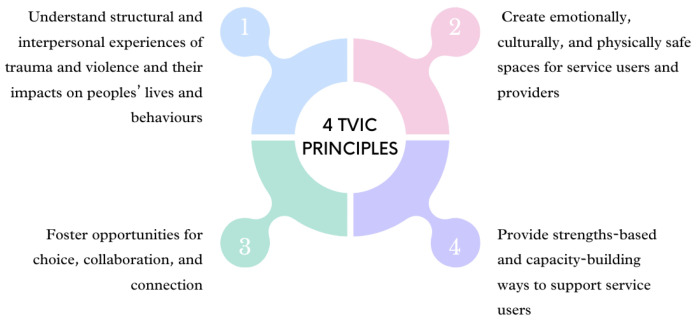
Four principles of trauma- and violence-informed care [13].

**Figure 2 ijerph-21-00254-f002:**
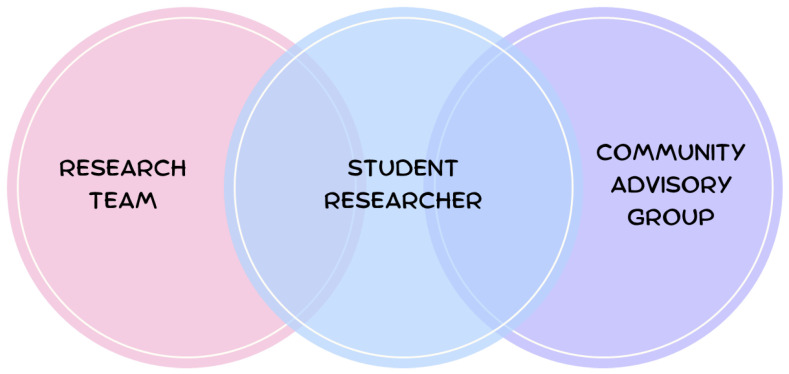
Research team structure and engagement with community.

## Data Availability

The data are unavailable due to ethical approval constraints. The participants were guaranteed that the raw data would remain confidential and not be disclosed.

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
