# Peer review of "Exploring Trauma- and Violence-Informed Pregnancy Care for Karen Women of Refugee Background: A Community-Based Participatory Study"

_ijerph, 2024, doi:10.3390/ijerph21030254_

Round 1
Reviewer 1 Report
Comments and Suggestions for Authors
I would like to start by applauding the main author, who is from a Karen refugee background, in conducting this study with much insight and cultural sensitivity. The paper seems to be a summary of an Honor's thesis, as indicated by the author(s), and it certainly stands apart as an Honor's thesis work on the basis of its authenticity, thoughtful methodology and good insights. I really liked the fact that a community group was involved in the construction of the research project and fieldwork, and having such bottom-up involvement fits with evolving standards of ethnographic fieldwork. I am also glad to see that the gendered concerns of maternity care for refugee populations is being discussed here within the framework of trauma-informed analysis, which is a unique contribution of the paper. I also like the Karen concepts being used to describe culturally-sensitive notions of comfortable and uncomfortable, indicating that our Western ways of defining 'good care' does not always match the care expectations of other communities.
Having said that, I have a few concerns to help upgrade the paper from an honor's thesis to scholarly article:
1) While I was very excited about the trauma-informed care framework, the materials presented in the findings do not touch on issues related to working with already traumatized communities. Being a refugee, even under the best conditions of displacement, is traumatizing. And the authors do a nice job of showing that we should try to not further traumatize these populations, especially vulnerable pregnant women. But the material does not show how prior trauma can inform the current experiences. Most of the statements quoted are fair points that any refugee or immigrant working with an interpreter in a hospital setting would say/have experienced. In fact, you don't need to be a refugee at all to make the same observations about confusion and lack of bedside manners. Thus, in the findings we have no sense that there is something bigger going on here except fair criticism of bedside manners of hospital staff and professionalism of the interpreters (who are usually community interpreters and not necessarily trained in all aspects of their jobs). The statements quoted and the little information provided on the Karen context do not do justice to the trauma previously experienced by these communities and how certain hardships can be compounded by hospital treatment of the patients.
2) There is literature out there that talks about post-settlement traumatization and that refugees expect hardship and trauma in their voyages but not after settlement in their new (industrialized) nations, and the post-settlement trauma can hit harder than the nightmares of refugee camps and dangerous journeys. This also falls along with the literature on the "grateful refugee" misconception, that refugee populations somehow should count their blessings that they are getting maternal care in developed-country hospitals, without understanding that their treatment in these settings is not necessarily superior, could leave much to be desired culturally, can be very difficult because of linguistic barriers to communication, and that the sterile interactions could be offensive, patronizing, confusing or re-traumatizing to women, especially. Given the focus of this paper on trauma-informed care, these points were not raised well in the current draft of the article.
3) This research clearly took place during Covid, and the restrictions of lockdown (such as inability to find in-person translators or being able to bring one's own family members for support and translation) and the anxieties of pregnancy during Covid need to be better articulated. Refugee populations were triple effected by transportation disruptions, employment problems and lack of access to non-profit help during Covid lockdowns. The traumatizing effects of this period should be emphasized as this paper is being published in a public health related issue of a journal, much more so than it currently is.
4) I would have liked to see a little more information on each women, since the study works with a handful of informants. Did they wish to conceive? Do they have any domestic abuse or sexual abuse experiences from the past? Did they have any miscarriages or problems with pregnancies? How did Covid contribute to their concerns? How many years had they spent in camps prior to being here? Where are they 5 years in? Education level differences or employment? Even age? I also would like to know if the communal group's involvement introduced restrictions to access? Meaning, given that the author is Karen, would more intimate details about the care situations come up in interviews if the community group had not been involved?
If the authors are able to strengthen the fieldwork and analysis sections to provide some responses to these questions, I think this paper has the potential to become a scholarly article. Thank you for the opportunity to read this paper!
Reviewer 2 Report
Comments and Suggestions for Authors
Dear Authors,
I feel honored to review this text. The topic is socially important, requires an ethical approach, and it would be good if the research resulted in applicable recommendations.
The article is clearly written, one might say too simple for a scientific article. There is a lot of content specific to educational books used by non-governmental organizations. This content is very important, but from the point of view of a scientific text, it is too extensive (in the first part of the article) and written in a language that explains the problem to students. Meanwhile, readers reading a research article will likely be quite familiar with the issue from a general perspective and will be interested in details specific to the Karen community in Australia.
The situation is similar in the part devoted to the organization of research. After all, we only have 7 respondents and one researcher. And the entire description of searching for respondents, qualifying them, and consulting with a team of advisors suggests much more research work. Here, the description of the study organization misbalanced the proportions. The description suggests terribly complicated, ethically confirmed procedures, but the result is a very minor study. The whole procedure is justified from the point of view of human rights (the most important!), but in the scientific text we see a clear dissonance between the scale of the statement and the research. This is my most serious objection to the text submitted.
I also have a few specific comments:
1. the article is to be read by readers from all over the world, so (e.g. page 3) geographical regions of Australia or city districts must be described in full form, understandable to people outside Melbourne/Australia.
2. Abstract - there is no mention of how small the study is and what location in Australia.
3. Sub-chapter 2.3.2 - sounds naive. Why sample questions (lines 240-248)? why not all? And how do they relate to the coding and titles of aspects according to which the content in Results is divided?
4. In the Results section, I would expect a greater selection of opinions from different respondents in the subsections. Meanwhile, we have one or two respondents each (with citations), which leaves us with a lack of knowledge whether other respondents would also say so...
5. Acknowledgments: I'm not sure if this is the correct way to write acknowledgments there.
6. In the text we quote literature like this [7]. dot at the end
7. It seems to me that the way literature is recorded in IJERPH (References) is slightly different.
I am convinced that the text needs to be corrected, made more scientific and published.
Comments on the Quality of English LanguageThe English language is understandable and clear, and the text contains only minor errors.
